# CREST: Cross-modal Resonance through Evidential Deep Learning for Enhanced Zero-Shot Learning

## ABSTRACT

Zero-shot learning (ZSL) enables the recognition of novel classes by leveraging semantic knowledge transfer from known to unknown categories. This knowledge, typically encapsulated in attribute descriptions, aids in identifying class-specific visual features, thus facilitating visual-semantic alignment and improving ZSL performance. However, real-world challenges such as distribution imbalances and attribute co-occurrence among instances often hinder the discernment of local variances in images, a problem exacerbated by the scarcity of fine-grained, region-specific attribute annotations. Moreover, the variability in visual presentation within categories can also skew attribute-category associations. In response, we propose a bidirectional cross-modal ZSL approach **CREST**. It begins by extracting representations for attribute and visual localization and employs Evidential Deep Learning (EDL) to measure underlying epistemic uncertainty, thereby enhancing the model's resilience against hard negatives. CREST incorporates dual learning pathways, focusing on both visual-category and attribute-category alignments, to ensure robust correlation between latent and observable spaces. Moreover, we introduce an uncertainty-informed cross-modal fusion technique to refine visual-attribute inference. Extensive experiments demonstrate our model's effectiveness and unique explainability across multiple datasets. Our code and data are available at: https://anonymous.4open.science/r/CREST-1CEC.

## CCS CONCEPTS

• **Computing methodologies → Learning paradigms**.

## KEYWORDS

Zero-Shot Learning, Multimodality, Evidential Deep Learning, Contrastive Learning

## 1 INTRODUCTION

Humans frequently possess the talent to grasp novel concepts relying on prior experience without the need to see them beforehand. For instance, a peacock is commonly known as a bird with a colorful fan-shaped tail; if individuals have previous knowledge of birds and fans, they can quickly identify a peacock. However, unlike humans, widely used and studied supervised deep learning models are typically limited to classifying samples belonging to categories seen during training, lacking the capacity to handle samples from unseen categories during training, thus lacking generality and flexibility. Therefore, to further advance Artificial General Intelligence (AGI) [4] and achieve true implementation, Zero-Shot Learning (ZSL) was introduced to identify new classes by leveraging inherent semantic relationships during learning [19, 33–35, 44]. It is already extensively applied in tasks with broad real-world applications, *e.g.*, image classification [18, 63], semantic segmentation [5, 17], video understanding [66, 71], 3D generation [29, 67], *etc.*, which

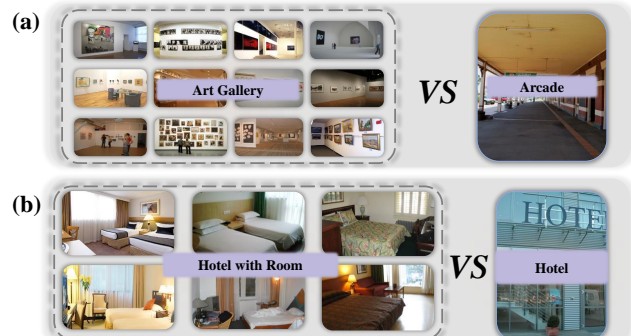

**Figure 1: Challenges in instance-level recognition in the real world: (a) Attributes distribution imbalances—significant frequency differences among attributes; (b) Attributes co-occurrence—tendency of certain attributes to appear together, influencing model bias (further statistical details are available in the Supplementary Material).**

also contributes significantly to the robust development of Large Language Models (LLMs) [31, 58] and Embodied AI [26, 56].

In ZSL, attributes stand as key semantic descriptors for visual features of images, representing a widely embraced form of annotation. Unfortunately, the attribute annotations are more often categorical rather than regional [13]. Dense attention interactions do not guarantee that models directly grasp the correspondence between local visual-semantic information and categories, nor do they alleviate the model's epistemic uncertainty when confronted with unseen categories [51]. That is because the skewed distribution of attributes in the real world, as well as the issues arising from attribute co-occurrence shown in Figure 1.

Existing methods overlook the importance of aligning regional features with categories. Models may link specific attributes, like a red bird's bill, to "bill color red" but struggle to deduce the bird's species. This challenge is compounded as attributes across species are often intertwined. Furthermore, real-world images of the same category vary significantly due to factors like camera angles, background, distances, lights and the motions, making it difficult for dense attention to learn hard category-matching patterns. This can increase epistemic uncertainty when merging features for inference, potentially exacerbating modal conflicts and impairing model performance [65].

To this end, we integrate Evidential Deep Learning (EDL) [51] into ZSL for the first time, leading to a novel framework, named Cross-modal Resonance through Evidential Deep Learning for Enhanced Zero-ShoT Learning, termed as CREST. Specifically, we employ the Visual Grounding Transformer (VGT) and the Attribute Grounding Transformer (AGT) to extract bidirectional, region-level features from images and attributes. Unlike conventional approaches that simply adjust distances within the representation

space based on category [13], our strategy addresses vision variability and feature-category alignment directly. We first introduce instance-level contrastive learning for adaptive vision alignment and employ a technique similar to non-maximum suppression to reduce attribute overlap between categories, facilitating deeper attribute-category insights. To counteract the potential degradation from hard-negative samples [48], we apply EDL for epistemic uncertainty measure and develop an uncertainty-driven fusion method [23, 24, 38, 65]. This enhances the model's generalization in downstream tasks by merging semantic knowledge across representation spaces. To summarize, our contributions are as follows:

- We propose CREST, a novel ZSL framework that considers bidirectional cross-modal representations of attributes and visual features. Moreover, it leverages dual learning pathways, focusing on both visual-category and attribute-category alignments, learning implicit matching patterns between features and categories from fine-grained visual elements and attribute texts.

- To the best of our knowledge, CREST is the first in ZSL to apply EDL for measuring epistemic uncertainty and mitigating potential conflicts in cross-modal fusion.

- Extensive experiments show that CREST performs competitively on three well-known ZSL benchmarks, i.e., CUB [59], SUN [46], and AWA2 [60]. Comprehensive ablations and analyses further validate the effectiveness and explainability of our approach.

## 2 RELATED WORK

### 2.1 Zero-shot learning

ZSL can be classified into two main categories based on the classes encountered during the testing phase: Conventional ZSL (CZSL) and Generalized ZSL (GZSL), where CZSL is designed to predict classes that have not been seen during training, whereas GZSL extends its predictive capability to both seen and unseen classes [60]. The core concept of ZSL revolves around learning discriminative and transferable visual features based on semantic information, e.g., attribute descriptions [34], sentence embeddings [49], and DNA [2], enabling effective visual-semantic interactions. Among these, attributes stand out as the most commonly used semantic information within ZSL. Early research focused on harnessing visual-semantic interactions to transfer knowledge to unseen categories [1, 37, 55]. These initial attempts, particularly through embedding-based methods, entailed learning a mapping between seen categories and their corresponding semantic vectors, followed by employing nearest neighbor searches within the embedding space to classify unseen categories [68, 70]. Since they primarily rely on seen category samples, the effectiveness was significantly limited due to a bias towards these categories, exacerbating the challenge in GZSL. Novel regularization and space modification strategies have been developed to improve ZSL model generalization [36, 45, 54]. Generative models, including VAEs [11, 12, 50, 57], GANs [10, 21, 61, 63], and generative flows [53, 74], synthetically enhance feature spaces with unseen class characteristics. These methods, aiming to bridge the domain gap, reframes ZSL as a supervised task by providing a means to compensate for the lack of unseen class data. Despite progress, these methods often neglect localized visual cues in favor of global information, overlooking the nuanced, fine-grained attributes essential for dissecting complex semantic categories [6, 14]. This oversight weakens the visual representations obtained, diminishing the efficacy of the visual-semantic knowledge transfer. Subsequently, intricate attentions are integrated into ZSL to prioritize salient features and attributes, improving model discernment [16, 28, 32, 39, 43, 75]. And Recent studies have started experimenting with the deployment of intricate attention to engage with region-level visual-attribute features [6–8, 13]. These methods highlight distinctive, fine-grained features, evolving towards complex attention modules for deeper semantic understanding. However, due to instance-level visual variability and inter-class attribute coupling, fine-grained representations may not guarantee accurate feature-to-category matching. This paper delves into aligning latent feature and category spaces.

### 2.2 Evidential Deep Learning for Classification

EDL enhances machine learning by enabling models to quantify uncertainty, thus bolstering reliability and interpretability. Grounded in subjective logic principles [30], EDL has emerged as a response to the challenges of model confidence and uncertainty, as highlighted in neural network calibration issues by [20]. The framework's utility was further solidified by [51], which introduced a method to quantify classification uncertainty, significantly increasing deep learning model trustworthiness. The adaptability of EDL to various data contexts has been demonstrated through applications like open set action recognition [3], signifying its efficacy in handling new and unseen data types. The scope of EDL further expanded to multi-view classification [23], showcasing its ability to integrate and reason with information from multiple sources. This integration was further enhanced by introducing dynamic evidential fusion [24], highlighting EDL's adaptability in complex data environments.

Recent advancements, such as adaptive EDL for semi-supervised learning [72] and its application in multimodal decision-making [52], have marked EDL's progression towards addressing real-world data challenges. Additionally, [65] illustrates EDL's potential in conflictive multi-view learning scenarios, reinforcing its capacity to support reliable decision-making across diverse applications. In ZSL, there exists epistemic uncertainty in the gap between region-level fine-grained latent space and category space. Moreover, dual-stream visual-attribute interactions do not necessarily align representation spaces. Therefore, we apply EDL to assess feature-category alignment uncertainties independently and introduces an uncertainty-driven fusion framework for coherent visual-attribute inference.

## 3 METHODOLOGY

### 3.1 Problem Definition

ZSL equips models to recognize targets in unseen categories. The training set, $D^s = \{(x^s, y^s)|x^s \in \mathcal{X}^s, y^s \in \mathcal{Y}^s\}$, consists of samples from known categories, with $x^s$ as images labeled $y^s$. The set $D^u = \{(x^u, y^u)|x^u \in \mathcal{X}^u, y^u \in \mathcal{Y}^u\}$ captures samples from new categories. With $\mathcal{Y}^u$ and $\mathcal{Y}^s$ distinct, each $y$ aligns with a category $c \in C = C^s \cup C^u$. This framework leverages attribute information from $C^s$ for knowledge transfer to $C^u$. Assuming predefined attributes for each category, quantified as either continuous or binary values, the dataset's attribute space is $\mathcal{A} = \{a_1, \ldots, a_{|\mathcal{A}|}\}$. Each

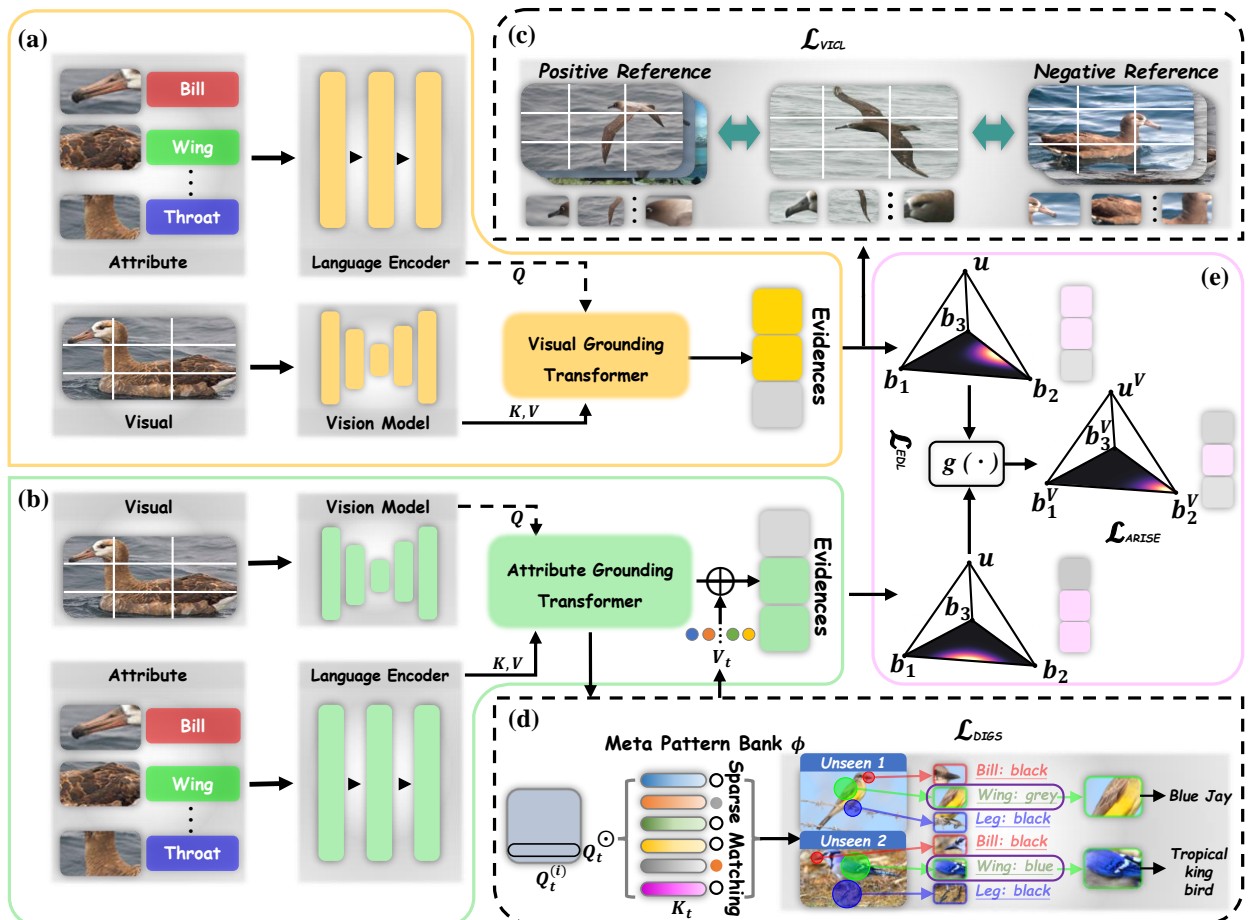

**Figure 2: The CREST model's architecture is depicted in Figure 2, initiating with modules (a) and (b) that perform bidirectional grounding to localize features within visuals and attributes. Following this, modules (c) and (d) engage in dual learning to align visual-category and attribute-category in the latent space. The process concludes with an uncertainty-driven fusion module (e), which integrates bidirectional evidence to enable robust visual-attribute inference.**

category's attribute profile, $c$, is depicted by $z^c = [z_1^c, \ldots, z_{|\mathcal{A}|}^c]^\top$, reflecting the value of each associated attribute.

## 3.2 Cross-modal Feature Extraction

**Feature Extraction: Attributes and Vision.** We extract textual features using the pre-trained GloVe model[47], while employing ResNet-101[25] as the CNN backbone to distill visual features from images (as depicted in Figure 2(a)(b)). These features support the development of a bidirectional grounding Transformer.

**Bidirectional Grounding Transformer.** In the decoding phase, the VGT and AGT refine visual and semantic attributes, respectively. The VGT attend semantic features to localize relevant image regions, whereas the AGT interprets semantic information through regional visual features. Both decoders employ a streamlined cross-attention module, with the encoder output $U$ serving as keys $K$ and values $V$, and semantic embeddings as queries $Q$. This methodology establishes a bidirectional link between images and attributes,

enhancing the recognition of unseen categories. The process is concisely described as follows:

$$K = UW_k, \quad V = UW_v, \quad Q = VW_q,$$

$$\hat{F} = \text{SoftMax}\left(\frac{QK^\top}{\sqrt{d_k}}\right)V, \tag{1}$$

where $W_q$, $W_k$, $W_v$ are the learnable weights in cross attention, $d_k$ represents the dimension of the features. After $n$ layers of iteration, the output $\hat{F}$ is transformed by a FeedForward layer:

$$F^V = \text{ReLU}\left(\left(\hat{F}W_1 + b_1\right)W_2 + b_2\right). \tag{2}$$

The AGT structure is analogous to the VGT, differing only in the modalities employed as queries in the cross-attention modules. Overall, the features of attribute and visual localization $F^A$, $F^V$ can be respectively captured through the application of AGT and VGT.

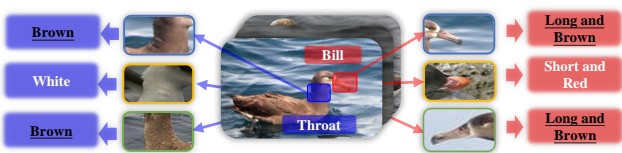

**Figure 4: Illustration of attribute coupling across bird species, highlighting shared and divergent traits.**

module $\Phi \in \mathbb{R}^{\phi \times d}$, where $\phi$ and $d$ ($d < |\mathcal{A}|$) respectively represents the total number of memory pattern vectors and their dimensional attributes.

$$Q^{(i)} = F_i^A W_Q + b_Q,$$
$$a_j^{(i)} = \frac{\exp(Q^{(i)}\Phi[j]^\top)}{\sum_{k=1}^{\phi} \exp(Q^{(i)}\Phi[k]^\top)}, \quad (5)$$
$$V_*^{(i)} = \sum_{j=1}^{\phi} a_j^{(i)} \Phi[j].$$

Specifically, in a batch with $N$ samples, our model uses the AGT to map the $h$-dimensional features $F^A \in \mathbb{R}^{N \times h}$ to the queries $Q \in \mathbb{R}^{N \times d}$ in the latent space of a meta pattern bank with $W_Q \in \mathbb{R}^{h \times d}$ and bias $b_Q \in \mathbb{R}^{h \times d}$. These queries compute similarity scores with pattern vectors $\Phi$ via dot products. Equation 5 delineates the transformation where $Q^{(i)} = F_i^A W_Q + b_Q$ generates the attention score $a_j^{(i)}$ that leads to the sparse attention-weighted feature vector $V_*^{(i)}$. This vector is subsequently remapped to the latent space of $F^A$, and directly added to it, enhancing the feature set by integrating the weighted information from the latent space. To decouple the attribute-category mapping in this latent space, we embrace the DIGS loss inspired by non-maximum suppression (NMS). It operates on two fronts:

*(i)*. The triplet loss component incentivizes the distinction between the closest and second-closest memory pattern vectors. Let $Q^{(i)}$ be the query representation for the $i$-th example, $\Phi[p]$ the most similar memory pattern (positive sample), and $\Phi[n]$ the second most similar memory pattern (negative sample). The triplet loss is then defined as:

$$\mathcal{L}_{\text{tp}} = \sum_{i=1}^{N} \max \left( \left\| Q^{(i)} - \Phi[p] \right\|^2 - \left\| Q^{(i)} - \Phi[n] \right\|^2 + \lambda, 0 \right), \quad (6)$$

where $\lambda$ is a margin enforcing that the similarity between $Q^{(i)}$ and $\Phi[p]$ exceeds that between $Q^{(i)}$ and $\Phi[n]$ by at least $\lambda$, encouraging the model to focus on positive samples and hard negatives and pull positive samples closer to the anchor $Q^{(i)}$ than negative ones.

*(ii)*. The regularization term promotes compact clustering of patterns by minimizing the distance between each query and its most similar memory pattern. This is quantified as:

$$\mathcal{L}_{\text{reg}} = \sum_{i=1}^{N} \left\| Q^{(i)} - \Phi[p] \right\|^2, \quad (7)$$

By synthesizing these components, the DIGS loss is articulated as $\mathcal{L}_{\text{DIGS}} = \mathcal{L}_{\text{tp}} + \mathcal{L}_{\text{reg}}$. Hence, it ensures that the memory patterns not only cluster tightly but also maintain separation, enabling the model to discern and generalize known patterns effectively while grasping the relational structure of the prototypes.

---

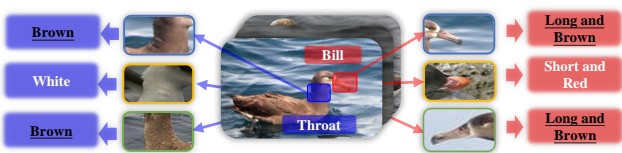

**Figure 3: The Birds of an identical category (*i.e.* black-footed albatross) captured in varying angles, backgrounds, distances, illumination, motions, *etc.* illustrating the dynamic nature of vision variability.**

## 3.3 Vision Instance-level Contrastive Learning

Generally speaking, existing methods achieve implicit alignment with the categorical space by mapping latent semantic matches in text to relevant visual regions in images, subsequently employing fine-grained embeddings. However, in the real world, the images captured often exhibit visual variability due to factors such as angles, backgrounds, distances, illumination, and motion (as shown in Figure 3). This variability significantly diminishes the practical effectiveness of textual semantics since the fine-grained visual representations derived from text may not necessarily correspond to the typical categories intended. Conversely, subjects from different categories might appear visually similar due to these influencing factors. To foster proximity among similar entities and distance among distinct categories in the representational space, some approaches might consider employing intra-group category labels for supervision. This method, however, could yield suboptimal solutions due to the vision variability present in an open-world scenario. To this end, we propose the Visual Instance-level Contrastive Learning (VICL) to mitigate the gap between fine-grained visual latent space and intra-category space.

$$\mathcal{L}_{\text{VICL}} = \mathbb{E}_{x \sim \mathcal{X}^s} \left[ -\log f_\theta(\tilde{v} \mid s, x) \right], \quad (3)$$
$$f_\theta = \frac{\exp \left( D\left( \tilde{v}, \tilde{v}^+ \right) / \tau \right)}{\exp \left( D\left( \tilde{v}, \tilde{v}^+ \right) / \tau \right) + \sum_{\tilde{v}^- \in \mathcal{N}(\tilde{v})} \exp \left( D\left( \tilde{v}, \tilde{v}^- \right) / \tau \right)}, \quad (4)$$

where $\tilde{v}$, $\tilde{v}^+$ and $\tilde{v}^-$ represent a candidate positive sample, its positive sample and negative sample respectively. And we adopt a strategy that adjusts for intra-category visual variability through a similarity-based selection of positive samples. Given a batch, the similarity score $D(\tilde{v}, \tilde{v}^+)$ is computed. If no intra-category sample resemble the candidate, we then identify the similar samples across the batch to serve as the positive samples, irrespective of category, based on the similarity score. This approach enables the model to maintain category coherence despite visual discrepancies.

## 3.4 Decoupled Insight for Grounding Semantics

Traditional methods typically align attribute features with visual features in a straightforward manner to achieve recognition outcomes. However, as illustrated in Figure 4, where attributes coupling across categories, posing challenges to accurate identification. As shown in Figure 2(d), similar visual regions can share the same attributes and intensify the challenges. Hence, we propose Decoupled Insight for Grounding Semantics (DIGS) loss and leverage a Meta-Pattern Bank to develop an auxiliary sparse attention

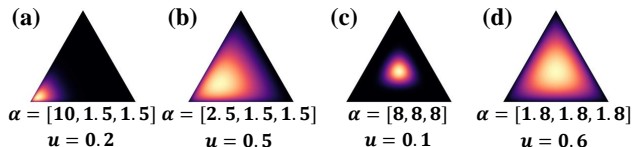

$$\alpha = [10, 1.5, 1.5] \quad \alpha = [2.5, 1.5, 1.5] \quad \alpha = [8, 8, 8] \quad \alpha = [1.8, 1.8, 1.8]$$
$$u = 0.2 \qquad\qquad u = 0.5 \qquad\qquad u = 0.1 \qquad\qquad u = 0.6$$

**Figure 5: Visualization of Classification Confidence. In a three-category classification context, the correct outcome is presumed to be the first category. Ideally, a model with good calibration should yield Confident and Precise (CP) decisions (a) or Erroneous and Uncertain (EU) outcomes (d). On the other hand, instances of Confident but Unclear (CU) judgments (b) and Erroneous but Positive (EP) assertions (c) are indicative of areas where model certainty needs to be aligned more accurately with its precision.**

## 3.5 Evidential deep learning

Given two opinions on the same instance, $\omega_A = (b^A, u^A, a^A)$ and $\omega_B = (b^B, u^B, a^B)$, their synthesis $\omega^{A \oplus B}$ combines their beliefs, uncertainty, and evidence as follows:

$$b_k^{A \oplus B} = \frac{b_k^A u^B + b_k^B u^A}{u^A + u^B}, \quad u^{A \oplus B} = \frac{2u^A u^B}{u^A + u^B}, \quad a_k^{A \oplus B} = \frac{a_k^A + a_k^B}{2},$$

where $a^A, a^B$ represent two different base distribution(*e.g.* Uniform distribution). The conflict degree $c(\omega^A, \omega^B)$ assesses the divergence and shared certainty between $\omega^A$ and $\omega^B$:

$$c(\omega^A, \omega^B) = c_p(\omega^A, \omega^B) \cdot c_c(\omega^A, \omega^B), \tag{8}$$

$$c_p(\omega^A, \omega^B) = 1/2 \sum_{k=1}^{K} |p_k^A - p_k^B|, \tag{9}$$

$$c_c(\omega^A, \omega^B) = (1 - u^A)(1 - u^B), \tag{10}$$

where $p$ represent the linear projected probability distributions of the opinions by Dirichlet parameters (*i.e.* $b$ and $u$). This framework facilitates a nuanced analysis of agreement and discord between the opinions.

As illustrated in Figure(2), we treat the outputs of VGT and AGT as evidence vectors, which typically involve issues of ambiguous recognition. Employing EDL allows us to precisely quantify these uncertainties, thereby deriving accurate recognition results. For each instance $\{\mathbf{x}_n^m\}_{m=1}^M$, the modality count $M$ encapsulates two modalities in our bidirectional grounding Transformer, namely visual-to-attribute and attribute-to-visual. The network computes Dirichlet distribution parameters $\boldsymbol{\alpha}_n^m = \mathbf{e}_n^m + 1$, where $\mathbf{e}_n^m = f_\theta^m(\mathbf{x}_n^m)$ is the predicted evidence vector, with $f_\theta^m$ denoting the modality-specific transformation function. The uncertainty mass derived as $u_n^m = \frac{K}{\sum_{k=1}^K (\alpha_{k,n}^m)}$, where $K = |C|$. Adapting to unimodal evidence-based classification, the traditional cross-entropy loss is intricately tailored for compatibility with this framework:

$$\mathcal{L}_{ACE}(\boldsymbol{\alpha}_n^m) = \int \left[ \sum_{j=1}^K -y_{nj} \log p_{nj} \right] \frac{\prod_{j=1}^K p_{nj}^{\alpha_{nj}^m - 1}}{B(\boldsymbol{\alpha}_n)} d\mathbf{p}_n, \tag{11}$$
$$= \sum_{j=1}^K y_{nj} \left( \psi(S_n) - \psi(\alpha_{nj}^m) \right),$$

where $\mathcal{L}_{ACE}(\boldsymbol{\alpha}_n^m)$ denotes the unimodal adaptive cross-entropy loss for the parameters $\boldsymbol{\alpha}_n^m$ of the Dirichlet distribution for a single

instance $n$. Utilizing the digamma function $\psi$, the integral is simplified to the expectation of the logarithm of predicted probabilities, where $S_n$ represents the sum of Dirichlet parameters for instance $n$, reflecting the total evidence across all classes. The objective of this adaptive loss function is to adjust the network's output parameters to accurately represent the inherent uncertainty in predictions, enabling the network to make confident predictions when evidence is ample and maintain a degree of uncertainty when evidence is scarce.

Nevertheless, the aforementioned loss function fails to address the issue of insufficient evidence caused by incorrect labels. Therefore, we incorporate a Kullback-Leibler (KL) divergence term into the loss function.

$$\mathcal{L}_{KL}(\boldsymbol{\alpha}_n^m) = KL\left[ D(\boldsymbol{p}_n \mid \tilde{\boldsymbol{\alpha}}_n^m) \| D(\boldsymbol{p}_n \mid 1) \right]$$
$$= \log\left( \frac{\Gamma\left( \sum_{k=1}^K \tilde{\alpha}_{nk}^m \right)}{\Gamma(K) \prod_{k=1}^K \Gamma(\tilde{\alpha}_{nk}^m)} \right), \tag{12}$$
$$+ \sum_{k=1}^K \left( \tilde{\alpha}_{nk}^m - 1 \right) \left[ \psi(\tilde{\alpha}_{nk}^m) - \psi\left( \sum_{j=1}^K \tilde{\alpha}_{nj}^m \right) \right],$$

where $D(\boldsymbol{p}_n \mid 1)$ represents the uniform Dirichlet distribution, $\tilde{\boldsymbol{\alpha}}_n^m = \mathbf{y_n} + (1 - \mathbf{y_n}) \odot \boldsymbol{\alpha}_n^m$ denotes the Dirichlet parameters after excluding non-misleading evidence from the predicted parameters $\boldsymbol{\alpha}_n^m$ for the $n$-th instance, and $\Gamma(\cdot)$ signifies the gamma function.

Hence, for the $n$-th instance in the single-modality setting with Dirichlet distribution parameter $\boldsymbol{\alpha}_n^m$, the loss is computed as follows:

$$\mathcal{L}_{ACC}(\boldsymbol{\alpha}_n^m) = L_{ACE}(\boldsymbol{\alpha}_n^m) + \lambda_t L_{KL}(\boldsymbol{\alpha}_n^m), \tag{13}$$

Where $\lambda_t = \min(1.0, t/\mathcal{E}) \in [0, 1]$ denotes the annealing coefficient, with $t$ being the index of the current training epoch and $\mathcal{E}$ representing the annealing steps. Gradually increasing the influence of KL divergence in the loss function prevents premature convergence of misclassified instances to a uniform distribution.

To ensure consistency across differing perspectives during training, a method to minimize the degree of opinion conflict is employed. The consistency loss for instance $\{\mathbf{x}_n^m\}_{m=1}^M$ is calculated as follows:

$$\mathcal{L}_{CON} = \frac{1}{M-1} \sum_{p=1}^M \left( \sum_{q \neq p}^M c(\omega_n^p, \omega_n^q) \right). \tag{14}$$

In the processes of VGT and AGT, mismatches may arise, linking attribute features to incorrect visual parts, or the reverse. The parameter $c$ serves to measure the conflict level between two opinions, where $c = 0$ denotes a lack of conflict and $c = 1$ denotes direct opposition. For the specific instance $\{\mathbf{x}_n^m\}_{m=1}^M$, the overall EDL loss functions can be given as follows:

$$\mathcal{L}_{EDL} = \mathcal{L}_{ACC}(\hat{\boldsymbol{\alpha}}_n) + \beta \sum_{m=1}^M \mathcal{L}_{ACC}(\boldsymbol{\alpha}_n^m) + \gamma \mathcal{L}_{CON}. \tag{15}$$

where $\hat{\boldsymbol{\alpha}}_n$ shaped by the fusion of modalities driven by uncertainty $u_n^m$ (*e.g.*, the uncertainty-weighted average of modalities' $\boldsymbol{\alpha}_n^m$) calibrates the EDL loss relative to the observed conflict degree.

## 3.6 Model training and optimization strategies

**Attribute Reinforced Semantic Integration.** We introduce an Attribute ReInforced SEmantic Integration (ARISE) to improve model discrimination by embedding attribute information into

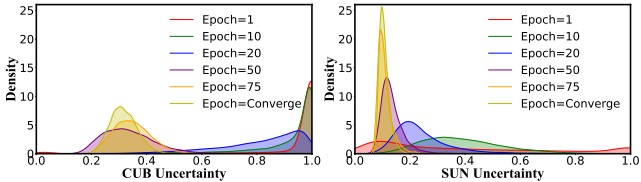

**Figure 6: Evolution of model uncertainty on CUB and SUN datasets with increasing epochs, showing a shift towards lower uncertainty as the model converges.**

the loss function, enhancing classification. By featuring a self-calibrating component, it mitigates overfitting and promotes attribute generalization, regulated by a balancing coefficient $\lambda_{CAL}$. Given a batch of $n_b$ training images $\{x_i\}_{i=1}^{n_b}$ with their corresponding class semantic vectors $z^c$, $\mathcal{L}_{ARISE}$ can be formally represented as follows:

$$\mathcal{L}_{ARISE} = -\frac{1}{n_b} \sum_{i=1}^{n_b} \left[ \log \frac{\exp\left(f\left(x_i\right) \cdot z^c\right)}{\sum_{\hat{c} \in C^s} \exp\left(f\left(x_i\right) \cdot z^{\hat{c}}\right)} \right.$$
$$\left. -\lambda_{CAL} \sum_{c'=1}^{C^u} \log \frac{\exp\left(f\left(x_i\right) \cdot z^{c'} + \mathbb{I}_{[c' \in C^u]}\right)}{\sum_{\hat{c} \in C} \exp\left(f\left(x_i\right) \cdot z^{\hat{c}} + \mathbb{I}_{[\hat{c} \in C^u]}\right)} \right] \tag{16}$$

where $f\left(x_i\right) = \mu \boldsymbol{\alpha}_i^A + (1-\mu) \boldsymbol{\alpha}_i^V$ with a blanced coefficient $\mu$. $\mathcal{L}_{ARISE}$ aims to minimize the discrepancy between the predicted and true distributions, taking into account the attribute similarities between categories, serving as a regularization term that encourages the model to learn generalizable features across different categories. Therefore, the overall loss can be obtained as follows:

$$\mathcal{L} = \mathcal{L}_{ARISE} + \mathcal{L}_{VICL} + \mathcal{L}_{DIGS} + \lambda_{EDL}\mathcal{L}_{EDL} \tag{17}$$

### 3.7 Zero-Shot Inference

Upon completing the training of CREST, we extract the visual embeddings of a test sample $x_i$ in the semantic space relative to VGT and AGT, denoted as $\boldsymbol{\alpha}_i^V$ and $\boldsymbol{\alpha}_i^A$. Given that the semantic-augmented visual embeddings from VGT and AGT offer complementary information, we integrate their predictions through combination coefficients $\mu$ for a calibrated test label prediction of $x_i$, expressed as:

$$c^* = \arg \max_{c \in C^u/C} \left(\mu \boldsymbol{\alpha}_i^A + (1-\mu) \boldsymbol{\alpha}_i^V\right)^\top \cdot z^c + \mathbb{I}_{[c \in C^u]} \tag{18}$$

In this formula, $C^u/C$ pertains to the CZSL/GZSL scenarios, respectively.

## 4 EXPERIMENTS

**Dataset.** Our study investigates three principal zero-shot learning (ZSL) benchmarks: two fine-grained datasets, CUB [59] and SUN [46], and one coarse-grained dataset, AWA2 [60]. CUB encompasses 11,788 images across 200 bird classes (150 seen, 50 unseen), featuring 312 attributes. SUN includes 14,340 images spanning 717 scene categories (645 seen, 72 unseen) with 102 attributes. AWA2 contains 37,322 images of 50 animal classes (40 seen, 10 unseen), each described by 85 attributes.

**Evaluation Protocols.** Following Xian et al.'s framework [62],

we evaluated the top-1 accuracy in both CZSL and GZSL setups. In CZSL, accuracy is assessed solely by predicting unseen classes. For GZSL, we compute the accuracy for both seen ($S$) and unseen ($U$) classes and employ their harmonic mean (defined as $H = (2 \times S \times U)/(S + U)$) as the evaluative metric.

**Implementation Details** We adopt the training divisions suggested by [61]. The feature extraction backbone is a ResNet101 architecture, which has been pre-trained on ImageNet and is utilized without further fine-tuning. The optimization is performed using the Adam optimizer, with hyperparameters set to learning rate of 0.0001 and a weight decay of 0.0001. And the batch size parameters is set to 64. Based on empirical evidence, the hyperparameters $\lambda_{EDL}$ and $\lambda_{CAL}$ are fixed at 0.001 and 0.2 across all datasets. Finally, the encoder and decoder layers of our bidirectional grounding Transformer are configured with a single attention head.

### 4.1 Comparison with the State of the Art

In our comparative analysis, we have examined 17 representative or state-of-the-art models from the period of 2020-2023, as illustrated in Table 1. Our CREST model consistently outperforms most models across the three benchmarks: CUB, SUN, and AWA2, in terms of CZSL accuracy. Notably, CREST achieves the highest harmonic mean (H) on both CUB and AWA2 benchmarks, indicating a well-balanced performance between seen (S) and unseen (U) classes, which is a critical measure in ZSL.

Our CREST model exhibits robust performance in the GZSL setting for unseen classes (U) on AWA2, achieving competitive accuracy. This highlights CREST's capability to recognize new categories effectively while maintaining strong performance on seen classes. Furthermore, the results indicate that while some models like TransZero++ [6] exhibit high accuracy in seen classes, they do not necessarily maintain this level of performance in unseen classes. In contrast, CREST delivers a more consistent and superior performance across both classes, emphasizing its efficacy in a more diverse and practical setting. The incremental advances observed with CREST affirm the effectiveness of our approach in addressing the challenges intrinsic to zero-shot learning, specifically in maintaining high discriminative power while effectively handling the domain shift between seen and unseen categories.

### 4.2 Ablation Studies

In the ablation study depicted in the first image, the effectiveness of various components of the CREST model is evaluated on CUB and SUN datasets. The study illustrates the importance of each component to the model's performance in both GZSL and CZSL.

The removal of the AGT from CREST results in a notable decrease in harmonic mean (H) and accuracy (ACC), demonstrating AGT's significant role in feature transformation. Without the VGT, the model's performance drops drastically, especially in the GZSL scenario, indicating VGT's critical contribution to visual feature integration. The exclusion of the EDL module also leads to diminished GZSL and CZSL outcomes, suggesting its key part in robust fusion and reinforcement of resilience against hard negatives.

Further analysis shows that the VICL and DIGS loss both enhance the GZSL performance, as their absence results in lower H scores. Setting the coefficient $\lambda_{CAL}$ of to zero slightly reduces the H scores

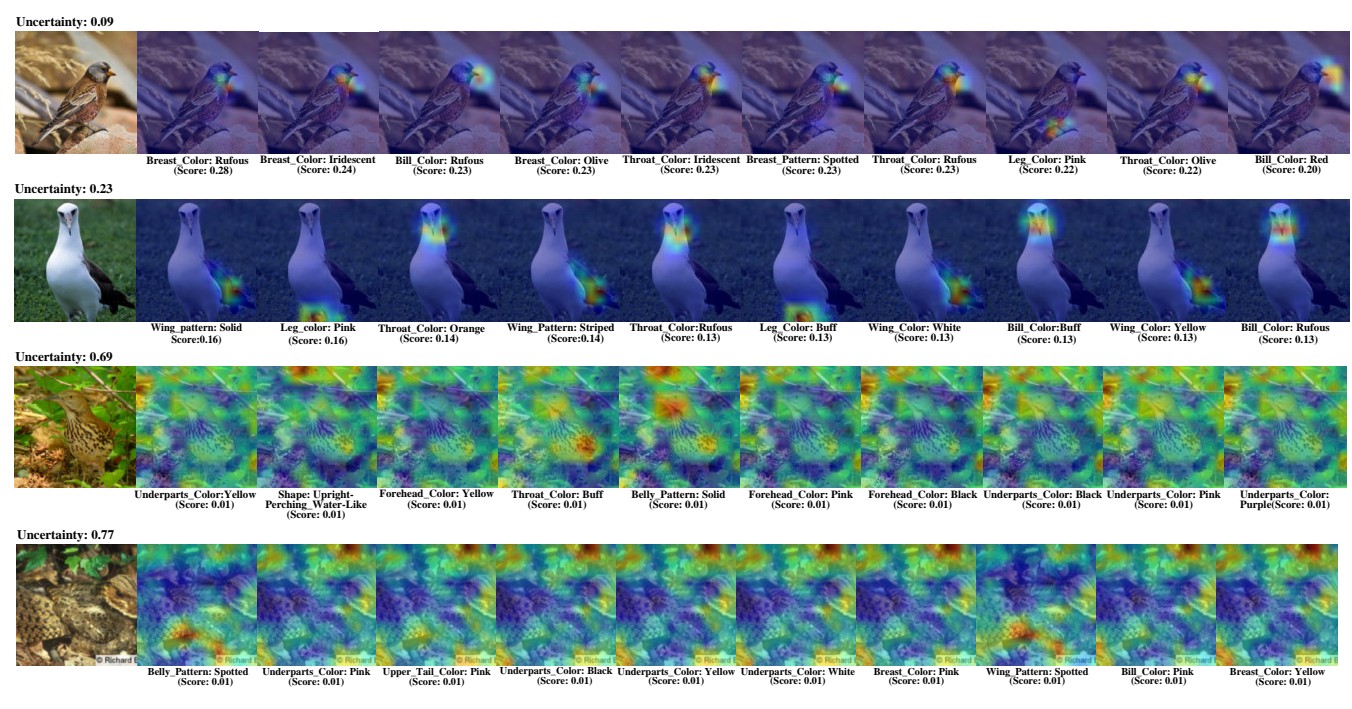

**Figure 7: Visualizing attention and uncertainty in attribute recognition on the CUB benchmark: Rows display attention maps for various bird species, with decreasing attribute certainty from top to bottom. Each image is annotated with attribute labels and corresponding confidence scores, highlighting the model's focus areas.**

**Table 1: Results(%) of CREST with the baselines on the CUB, SUN, and AWA2 benchmarks. Asterisks (\*) identify journal articles, while Underlined numbers denote second-highest results. And Bold figures highlight the leading metrics. Performance metrics encompass CZSL accuracy (ACC), GZSL accuracies for unseen (U) and seen (S) classes, and the harmonic mean (H) computed as $H = \frac{2 \times S \times U}{S + U}$, which gauges the equilibrium between U and S. ACC represents the top-1 classification accuracy in CZSL.**

| Methods | CUB | | | | SUN | | | | AWA2 | | | |
|---|---|---|---|---|---|---|---|---|---|---|---|---|
| | CZSL | GZSL | | | CZSL | GZSL | | | CZSL | GZSL | | |
| | ACC | U | S | H | ACC | U | S | H | ACC | U | S | H |
| TF-VAEGAN [42] (ECCV'20) | 64.9 | 52.8 | 64.7 | 58.1 | 66.0 | 45.6 | 40.7 | 43.0 | 72.2 | 59.8 | 75.1 | 66.6 |
| Composer [27] (NeurIPS'20) | 69.4 | 56.4 | 63.8 | 59.9 | 62.6 | 55.1 | 22.0 | 31.4 | 71.5 | 62.1 | 77.3 | 68.8 |
| APN [69] (NeurIPS'20) | 72.0 | 65.3 | 69.3 | 67.2 | 61.6 | 41.9 | 34.0 | 37.6 | 68.4 | 57.1 | 72.4 | 63.9 |
| DVBE [41] (CVPR'20) | - | 53.2 | 60.2 | 56.5 | - | 45.0 | 37.2 | 40.7 | - | 63.6 | 70.8 | 67.0 |
| DAZLE [28] (CVPR'20) | 66.0 | 56.7 | 59.6 | 58.1 | 59.4 | 52.3 | 24.3 | 33.2 | 67.9 | 60.3 | 75.7 | 67.1 |
| RGEN [64] (ECCV'20) | 76.1 | 60.0 | 73.5 | 66.1 | 63.8 | 44.0 | 31.7 | 36.8 | 73.6 | **67.1** | 76.5 | 71.5 |
| CE-GZSL [22] (CVPR'21) | 77.5 | 63.1 | 66.8 | 65.3 | 63.3 | 48.8 | 38.6 | 43.1 | 70.4 | 63.1 | 78.6 | 70.0 |
| GCM-CF [73] (CVPR'21) | - | 61.0 | 59.7 | 60.3 | - | 47.9 | 37.8 | 42.2 | - | 60.4 | 75.1 | 67.0 |
| FREE [10] (ICCV'21) | - | 55.7 | 59.9 | 57.7 | - | 47.4 | 37.2 | 41.7 | - | 60.4 | 75.4 | 67.1 |
| HSVA [11] (NeurIPS'21) | 62.8 | 52.7 | 58.3 | 55.3 | 63.8 | 48.6 | 39.0 | 43.3 | - | 59.3 | 76.6 | 66.8 |
| AGZSL [15] (ICLR'21) | 57.2 | 41.4 | 49.7 | 45.2 | 63.3 | 29.9 | 40.2 | 34.3 | **73.8** | 65.1 | 78.9 | 71.3 |
| GEM-ZSL [40] (CVPR'21) | 77.8 | 64.8 | 69.3 | 67.2 | 62.8 | 38.1 | 35.7 | 36.9 | 67.3 | 64.8 | 77.5 | 70.6 |
| MSDN [8] (CVPR'22) | 76.1 | 68.7 | 67.5 | 68.1 | 65.8 | 52.2 | 34.2 | 41.3 | 70.1 | 62.0 | 74.5 | 67.7 |
| TransZero [7] (AAAI'22) | 76.8 | 69.3 | 68.3 | 68.8 | 65.6 | 52.6 | 33.4 | 40.8 | 70.1 | 61.3 | 82.3 | 70.2 |
| TransZero++ [6] (TPAMI'22)* | 78.3 | 67.5 | **73.6** | 70.4 | **67.6** | 48.6 | 37.8 | 42.5 | 72.6 | 64.6 | 82.7 | 72.5 |
| DUET [13] (AAAI'23) | 72.3 | 62.9 | 72.8 | 67.5 | 64.4 | 45.7 | **45.8** | 45.8 | 69.9 | 63.7 | 84.7 | 72.7 |
| DSP [9] (ICML'23) | - | 62.5 | 73.1 | 67.4 | - | **57.7** | 41.3 | **48.1** | - | 63.7 | **88.8** | **74.2** |
| CREST (Ours) | **78.6** | **71.1** | 72.4 | **71.7** | 66.3 | 50.4 | 39.8 | 43.2 | 73.5 | 63.9 | 87.5 | 74.1 |

**Table 2: Ablation results for CREST on CUB and SUN datasets, detailing GZSL and CZSL performance for unseen (U) and seen classes (S), harmonic mean (H), and ACC.**

| Methods | CUB | | | | SUN | | | |
|---|---|---|---|---|---|---|---|---|
| | GZSL | | | CZSL | GZSL | | | CZSL |
| | U | S | H | ACC | U | S | H | ACC |
| CREST w/o AGT | 0.640 | 0.684 | 0.661 | 0.741 | 0.465 | 0.316 | 0.613 | 0.626 |
| CREST w/o VGT | 0.262 | 0.404 | 0.445 | 0.477 | 0.333 | 0.304 | 0.574 | 0.586 |
| CREST w/o EDL | 0.709 | 0.726 | 0.718 | 0.780 | 0.519 | 0.318 | 0.394 | 0.644 |
| CREST w/o VICL | 0.684 | 0.711 | 0.697 | 0.767 | 0.55 | 0.291 | 0.381 | 0.615 |
| CREST w/o DIGS | 0.689 | 0.722 | 0.705 | 0.769 | 0.468 | 0.326 | 0.385 | 0.606 |
| CREST $\lambda_{CAL}$ = 0 | 0.592 | 0.720 | 0.650 | 0.761 | 0.462 | 0.331 | 0.386 | 0.624 |
| CREST (Full) | 0.711 | 0.724 | 0.717 | 0.786 | 0.504 | 0.398 | 0.432 | 0.663 |

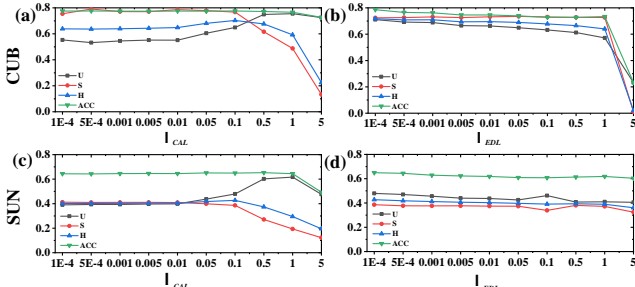

**Figure 8: Parameter tuning results for $\lambda_{CAL}$ and $\lambda_{EDL}$ of corresponding loss functions on the CUB and SUN datasets.**

but the overall full model displays superior performance in terms of both H and ACC, which solidifies the synergy and necessity of the full complement of CREST in achieving state-of-the-art results.

## 4.3 Hyperparameter Analysis

The parameter tuning for the CREST model indicates a clear optimum range for both $\lambda_{CAL}$ and $\lambda_{EDL}$. Performance peaks at moderate values of $\lambda_{CAL}$ before declining, signifying its critical role in balancing GZSL and CZSL outcomes. The influence of $\lambda_{EDL}$ appears more stable, with only a slight drop at high values, suggesting its robust contribution to the model's consistent performance across diverse visual tasks. These findings highlight CREST's ability to maintain accuracy while effectively generalizing to new categories, marking its strengths in a zero-shot learning context.

## 4.4 Qualitative Results

**Dynamic Uncertainty Progressive Reduction Visualizations.** Figure 6 showcases the evolution of model uncertainty for both the CUB and SUN datasets over training epochs. The density plots vividly demonstrate how uncertainty decreases as the epochs progress, with a significant shift towards lower uncertainty levels upon model convergence. This provides empirical evidence of CREST's learning stability and its increasing confidence in predicting class attributes over time, reflecting its robustness and efficacy in handling diverse data.

**Attention Mapping and Confidence Scoring Visualizations.** In Figure 7, attention visualization on the CUB Dataset is coupled with uncertainty quantification in attribute recognition. The

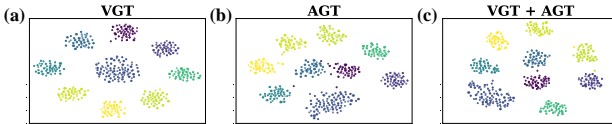

**Figure 9: t-SNE visualizations of features for classes in GZSL, with settings including a random selection of 10 classes from both seen and unseen categories. (a) and (b) illustrate the distinct clusters formed by VGT and AGT, respectively. Subfigure (c) displays the integrated representation post-EDL fusion, denoting the combined VGT and AGT spaces, which shows enhanced clustering of attributes across classes.**

descending order of rows from top to bottom corresponds to a decrease in attribute certainty, with each image annotated with attribute labels and scores. This not only confirmsCREST's nuanced understanding of attribute saliency but also illustrates the impact of real-world variables such as background clutter and occlusions on the model's performance. Additionally, the model demonstrates a keen perception of hard negatives, as reflected in higher uncertainty scores for attributes that are ambiguous or potentially misleading, which underscores its advanced capability for self-assessment and adaptability in complex visual scenarios.

**t-SNE Visualizations.** For Figure 9, the t-SNE visualizations illustrate the distinct clustering capabilities of the CREST model. The separate subfigures (a) and (b) highlight the feature spaces created by the VGT and AGT, respectively. Subfigure (c) reveals how the integration of VGT and AGT, through EDL fusion, enhances the distinctiveness of clusters, successfully separating the 10 randomly selected classes from both seen and unseen categories. This indicates CREST's powerful ability to delineate classes in a shared feature space, essential for ZSL.

## 5 CONCLUSION AND FUTURE WORK

In conclusion, CREST introduces a pioneering bidirectional cross-modal framework that adeptly addresses the visual-semantic gap in ZSL. By navigating distribution imbalances and attribute co-occurrence, it employs localized representation extraction and EDL-based uncertainty estimation, enhancing resilience and improving alignment in challenging ZSL scenarios. Our extensive evaluations establish CREST's advanced capabilities, reinforcing its standing as an effective and interpretable ZSL solution.

In future work, we intend to integrate CREST with LLMs to further enhance semantic alignment and interpretability. This integration aims to enrich CREST's robustness in handling complex zero-shot learning scenarios, thereby extending its applicability and effectiveness in ZSL tasks. Through this synergy, we seek to unlock deeper semantic insights and cater to a broader range of applications in the ever-evolving landscape of machine learning.

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
