# OpenReview forum: "CREST: Cross-modal Resonance through Evidential Deep Learning for Enhanced Zero-ShoT Learning"
_acmmm.org/ACMMM/2024/Conference — MM2024 Poster_

### Official Review · Reviewer_ek1r · 2024-05-22

**Rating:** 2
**Confidence:** 3

**Summary:**

The paper proposes a zero-shot learning framework called CREST that aims to enhance cross-modal visual-attribute alignment for recognizing unseen categories. CREST has three key components: 1) Cross-modal feature extraction using grounding transformers to extract localized visual and attribute representations. 2) Instance-level contrastive learning to align visual representations with categories, accounting for real-world visual variability. 3) Decoupled insight using a meta pattern bank and sparse attention to decouple attribute mappings across categories. It also introduces evidential deep learning for uncertainty measurement and uncertainty-driven visual-attribute fusion. The paper reports experimental results on three benchmark datasets.

**Strengths:**

1. The paper conducts experiments across three well-known zero-shot learning benchmark datasets (CUB, SUN, AWA2) which is adequate.
2. It uses grounding transformers to extract localized visual and attribute features, going beyond just global image/text features, which is theoretically well-motivated.
3. The formulations and descriptions of the various model components are technically correct and clear.

**Limitations:**

1. The overall novelty is limited: (1) It looks like an increment of the previous work [1], which presented an attribute→visual Transformer sub-net and a visual→attribute Transformer sub-net (VAT) to learn attribute-based visual features and visual-based attribute features. (2) The use of contrastive learning for aligning vision representations is not novel and has been studied in other works like CE-GZSL [2] (CVPR’21).
2. On the standard ZSL benchmarks CUB, SUN and AWA2, the performance gains over recent previous methods are relatively modest or lower than previous approaches. Considering the limited novelty, the improvement of the method is not significant.
3. The ablation studies could be more comprehensive to tease apart the contributions of different components. The Evidence Deep Learning consists of three important components, but according to Table 2, this paper only analyzes them as a group, and it is not feasible to clearly show the contribution of each component.
4. According to Table 2, The Evidence Deep Learning contributes marginally to performance improvement and there is no analysis for the different components in Evidence Deep Learning, so the novelty and contribution of Evidence Deep Learning in this paper is ambiguous.
5. According to Figure 9, it is not easy to distinguish the performance improvement that comes from the utilization of Evidence Deep Learning since the three visualization results are very similar.

[1] TransZero++: Cross Attribute-Guided Transformer for Zero-Shot Learning.
[2] Contrastive embedding for generalized zero-shot learning.

**Suitability:**

3

---

### Official Review · Reviewer_khPv · 2024-05-24

**Rating:** 4
**Confidence:** 4

**Summary:**

The authors propose a bidirectional cross-modal ZSL approach CREST. It begins by extracting representations for attribute and visual localization and employs Evidential Deep Learning (EDL) to measure underlying epistemic uncertainty, thereby enhancing the model’s resilience against hard negatives. CREST incorporates dual learning pathways, focusing on both visual-category and attribute-category alignments, to ensure robust correlation between latent and observable spaces. Moreover, the authors introduce an uncertainty-informed cross-modal fusion technique to refine visual-attribute inference.

**Strengths:**

Positive
- The authors proposed a new method targeting at athe zero-shot learning task.
- The authors conduct extensive experiments to demonstrate the effectiveness  of the proposed method.

**Limitations:**

Negative
- Besides the accuracy performance of the proposed method, the authors should compare the performance of the proposed method and the related methods.
- The motivation of the proposed model is unclear. For example, why did the authors set the the function g()?
- From Table 1, the performance of the proposed method is not significant on the datasets of SUN, and AWA2.

**Suitability:**

2

---

### Official Review · Reviewer_oJvV · 2024-05-24

**Rating:** 4
**Confidence:** 4

**Summary:**

This paper proposes a bidirectional cross-modal zero-shot learning framework to address the attribute distribution imbalance and co-occurence problems. The model is trained in dual pathways to perform visual-class and attribute-class alignments. In visual-class alignment module, the authors utilized contrastive learning to capture fine-grained visual features while keeping class coherence. In attribute-class alignment module, the authors utilized sparse matching and NMS-style loss function to decouple attribute-class mapping patterns. In addition, an EDL-based fusion module is proposed to quantify and fuse visual-attribute evidences. An additional regularization loss is also proposed to enhance the model’s discriminative ability. The experiments show promising results on popular ZSL datasets and effectiveness of the proposed modules.

**Strengths:**

- Well-motivated idea. The issues that this paper aims to address are practical and reasonable in real ZSL scenes. The motivation is well illustrated in figure 1,3,4.
- Solid modeling. The description of the three proposed modules is well formulated and I think it’s feasible to support the motivation. The introduction of EDL is novel comparing to other ZSL approaches.
- Good experimental results. This paper shows promising results on popular ZSL datasets. The ablation studies and visualizations show the effectiveness of proposed modules and loss functions.

**Limitations:**

- The results on SUN and AWA2 are relatively low comparing to other SOTAs.
- Some expressions are unclear, e.g., in Section 4.2, “In the ablation study depicted in the first image”, which image? Which table does this ablation refer to? In Section 4.3, the figure of results isn’t specified clearly yet. I recommend the authors to polish the paper carefully.
- Can the authors provide some complexity analysis of each module? It may require additional time and space to train the framework comparing to other approaches.
- Unfortunately, the code isn’t available on the provided link.

**Suitability:**

3

---

### Official Review · Reviewer_C8Yp · 2024-05-29

**Rating:** 4
**Confidence:** 3

**Summary:**

In this paper, the authors propose a novel bidirectional cross-modal ZSL approach CREST. CREST leverages dual learning pathways, focusing on both visual-category and attribute-category alignments. Furthermore, CREST applies Evidential Deep Learning (EDL) to measure underlying epistemic uncertainty and mitigate potential conflicts in cross-modal fusion.

**Strengths:**

- This paper is overall well-written and easy-to-follow.
- Numerous experiments show the effectiveness of the proposed method.

**Limitations:**

- EDL seems useless for the GZSL setting in CUB. When performing CREST w/o EDL, the metric H even becomes better. Could the authors explain this phenomenon?
- TransZero++[1] also introduce a dual-path learning by incorprating visual-to-semantic and semantic-to-visual mappings based on Transformer. Could the authors emphasize any difference between TransZero++ and the proposed method?

[1] TransZero++: Cross Attribute-guided Transformer for Zero-Shot Learning. T-PAMI 2022.

**Suitability:**

3

---

### Meta-Review · Area_Chair_dURQ · 2024-06-28

**Recommendation:** Accept (Poster)
**Confidence:** 5

**Metareview:**

Initially, the paper received one Weak Reject and three Borderline Accept scores. The reviewers acknowledged the good motivation and effectiveness of the proposed CREST approach but raised major concerns about the contribution and the similarity to existing methods like TransZero++, the performance on certain datasets, and the clarity and completeness of the experimental results.

The rebuttal addressed these concerns by providing clarifications and additional insights. After the rebuttal, Reviewer ek1r raised the score from  Weak Reject to Borderline Accept, and the other reviewers maintained their Borderline Accept scores, confirming that their concerns were addressed.

The AC has carefully read the paper, reviews, and rebuttal. The AC found the arguments of Reviewers oJvV and khPv to be the most persuasive and weighted the paper's strengths over its current weaknesses - while the experiments and expression can be further improved, its good motivation and solid modeling have potential implications for future research in related areas. The authors are encouraged to improve the final version by incorporating the reviews into the final version.